# Acetylated α-Tubulin and α-Synuclein: Physiological Interplay and Contribution to α-Synuclein Oligomerization

**DOI:** 10.3390/ijms241512287

**Published:** 2023-07-31

**Authors:** Alessandra Maria Calogero, Milo Jarno Basellini, Huseyin Berkcan Isilgan, Francesca Longhena, Arianna Bellucci, Samanta Mazzetti, Chiara Rolando, Gianni Pezzoli, Graziella Cappelletti

**Affiliations:** 1Department of Biosciences, Università degli Studi di Milano, 20133 Milan, Italy; milo.basellini@unimi.it (M.J.B.); huseyin.isilgan@unimi.it (H.B.I.); samanta.mazzetti@gmail.com (S.M.); chiara.rolando@unimi.it (C.R.); 2Fondazione Grigioni per il Morbo di Parkinson, 20125 Milan, Italy; pezzoli@parkinson.it; 3Department of Molecular and Translational Medicine, University of Brescia, 25123 Brescia, Italy; f.longhena@unibs.it (F.L.); arianna.bellucci@unibs.it (A.B.); 4Parkinson Institute, ASST-Pini-CTO, 20126 Milan, Italy; 5Center of Excellence on Neurodegenerative Diseases, Università degli Studi di Milano, 20133 Milan, Italy

**Keywords:** α-synuclein, acetylated α-tubulin, Tubacin, Parkinson’s disease, neurodegeneration, oligomers, Proximity Ligation Assay

## Abstract

Emerging evidence supports that altered α-tubulin acetylation occurs in Parkinson’s disease (PD), a neurodegenerative disorder characterized by the deposition of α-synuclein fibrillary aggregates within Lewy bodies and nigrostriatal neuron degeneration. Nevertheless, studies addressing the interplay between α-tubulin acetylation and α-synuclein are lacking. Here, we investigated the relationship between α-synuclein and microtubules in primary midbrain murine neurons and the substantia nigra of post-mortem human brains. Taking advantage of immunofluorescence and Proximity Ligation Assay (PLA), a method allowing us to visualize protein–protein interactions in situ, combined with confocal and super-resolution microscopy, we found that α-synuclein and acetylated α-tubulin colocalized and were in close proximity. Next, we employed an α-synuclein overexpressing cellular model and tested the role of α-tubulin acetylation in α-synuclein oligomer formation. We used the α-tubulin deacetylase HDAC6 inhibitor Tubacin to modulate α-tubulin acetylation, and we evaluated the presence of α-synuclein oligomers by PLA. We found that the increase in acetylated α-tubulin significantly induced α-synuclein oligomerization. In conclusion, we unraveled the link between acetylated α-tubulin and α-synuclein and demonstrated that α-tubulin acetylation could trigger the early step of α-synuclein aggregation. These data suggest that the proper regulation of α-tubulin acetylation might be considered a therapeutic strategy to take on PD.

## 1. Introduction

Tubulin is emerging as one of the interactors of α-synuclein, a small naturally unfolded protein expressed mainly in neurons. Due to its prevalent localization at the pre-synapses [1,2], the majority of studies on α-synuclein have focused on deciphering its role in this compartment [3]. Nevertheless, new additional roles for α-synuclein are emerging, including its involvement in microtubule cytoskeleton regulation and functions [4]. Starting from the ability of α-synuclein to co-immunoprecipitate with α- and β-tubulin [5], it has been proposed that α-synuclein not only modulates microtubule assembly and dynamics [6,7,8,9] but also axonal transport [10] and vesicle endocytosis [11]. Interestingly, α-synuclein was even found to be associated with tubulin at pre-synapse [12], hinting that tubulin/α-synuclein interplay may also play a relevant role in the control of synaptic homeostasis and neurotransmission.

α-synuclein is very well known as the main component of Lewy bodies [13], insoluble fibrillary deposits that, together with the loss of nigrostriatal dopaminergic neurons, are the key neuropathological hallmarks of Parkinson’s disease (PD). Recent studies have revealed that these proteinaceous aggregates are composed by organelles, vesicles, and different proteins, including cytoskeletal proteins [14]. Interestingly, Lewy bodies include not only tubulin [6,15,16] but also the histone deacetylase HDAC6 and Phospho-HDAC6 [17,18], the main microtubule-associated deacetylase responsible for α-tubulin deacetylation [19]. This is of great interest and suggests that the regulation of acetylated α-tubulin is altered during neurodegeneration.

Acetylation of α-tubulin on Lysine 40 (K40) is one of the most investigated post-translational modifications of the microtubule cytoskeleton. Historically associated with long-lived stable microtubules, the acetylation of α-tubulin is not merely a marker for microtubule stability but has been directly involved in the acquisition of physical properties of microtubules. Indeed, the acetylation of K40 alters the conformation of the flexible loop containing this residue [20], thus restricting its motion, weakening lateral contacts, enhancing microtubule flexibility, and making microtubules more resistant to mechanical stresses and breakages [21,22]. On the contrary, deacetylation increases the lateral contacts, making microtubules inflexible and susceptible to stress [21]. In neurons, the acetylation of α-tubulin has been mainly linked to the regulation of morphogenesis, neuronal differentiation, and neuronal transport [23].

A defective regulation of tubulin acetylation emerged to be linked to α-synuclein pathology in cellular and animal models [24]. Indeed, the treatment with the PD-related toxin MPP^+^ led to the increase of acetylated α-tubulin in neuronal-like cells [25] and to the fragmentation of acetylated microtubules and mitochondrial axonal transport impairment in mouse mesencephalic neurons [26]. Interestingly, a treatment with the environmental PD-related toxin 2,5-hexanedione did not affect the amount but the redistribution of acetylated α-tubulin from the neurites to the cell body [27]. An overall increase in acetylation of α-tubulin was also detected in dopaminergic terminals of the corpus striatum and in cell bodies of dopaminergic neurons in the substantia nigra of mice injected with the neurotoxin MPTP [28]. Increased tubulin acetylation was found in embryonic fibroblasts derived from *LRRK2* knockout mice [29] and skin fibroblasts isolated from patients harboring *LRRK2* mutation [30]. On the contrary, the decrease of acetylated microtubules and the disruption of axonal transport were observed in rat cortical neuron cultures expressing pathogenic forms of LRRK2 [31]. Furthermore, tubulin acetylation was also altered in *PARK2* knockout mice and in human neurons from patients harboring *PARK2* mutations [32]. Finally, a decrease in acetylated α-tubulin levels in the substantia nigra pars compacta was observed by the western blotting analyses of post-mortem human brain samples obtained from PD patients (including Braak stages IV–VI), but not in the hippocampus nor in the temporal cortex [33].

Despite the evidence pointing to a correlation between acetylated α-tubulin and α-synuclein, there are still many open questions, including whether they interact in a physiological context and the possible role of this interaction in triggering α-synuclein aggregation. Here, we investigated the interplay between acetylated α-tubulin and α-synuclein in mouse primary neurons and in post-mortem human brains. Our results reveal the physiological colocalization of the two proteins along microtubules and, moreover, highlight their physical close proximity in the substantia nigra. Finally, we modulated the acetylation of α-tubulin in a neuronal-like model and unraveled its impact on triggering the aggregation of α-synuclein.

## 2. Results

### 2.1. Endogenous α-Synuclein Colocalizes with Acetylated Microtubules in Primary Neurons

We first investigated the localization of α-synuclein and acetylated α-tubulin in primary mouse midbrain neurons. We isolated primary neurons from mouse embryo at 12.5 days of gestation (E12.5) and cultured them in vitro for 7 days (Figure 1A). Acetylated α-tubulin was detectable along microtubules mainly in the axon and in the neurites. On the contrary, α-synuclein was predominantly present in the cell bodies, despite being also detectable along neurites with a point signal. The high fraction of α-synuclein colocalizing with acetylated α-tubulin was pointed out by the Manders coefficient (M1 = 0.798 ± 0.063, Appendix A). On the other hand, a smaller fraction of acetylated α-tubulin colocalized with α-synuclein (M2 = 0.636 ± 0.101, Appendix A), supporting that a subset of the acetylated microtubule cytoskeleton may interact with α-synuclein. 

Taking advantage of Structured Illumination Microscopy (SIM), which allows for a higher resolution, we deeply investigated the α-tubulin/α-synuclein interplay. As shown in Figure 1B,C, we observed the localization of α-synuclein along single acetylated microtubules inside a bundle. The orthogonal view of the image supported the colocalization of the two signals (yellow spots), as highlighted in the YZ and XZ planes. As reported by the intensity profiles of the two different spectra along a single microtubule, α-synuclein did not always overlap with acetylated α-tubulin (Figure 1(Da)), confirming the partial colocalization of the two proteins. In particular, we found that the α-synuclein signal was also detectable in the space between the acetylated microtubules that ran parallel to each other (Figure 1(Db)).

Altogether, these results indicate that α-synuclein and acetylated α-tubulin partially colocalize in primary neurons, revealing the interplay of the two proteins in a physiological context. 

### 2.2. Acetylated α-Tubulin and α-Synuclein Colocalize in Post-Mortem Human Brain

To identify the relationship of acetylated α-tubulin and α-synuclein in post-mortem human brains, we investigated their localization in the substantia nigra. As shown in Figure 2, microtubules were strongly positive for acetylated α-tubulin. Regarding α-synuclein, two different signals were detectable: the first one was bigger and more intense (arrowheads, Figure 2B) than the second one, which was small and faint (arrows, Figure 2B). Whereas the first signal appeared to be near and almost leaned on microtubules, recalling synaptic punctate staining, the second seemed to localize with acetylated α-tubulin. The orthogonal view showed the colocalization of these small, faint α-synuclein signals with acetylated α-tubulin along microtubules (Figure 2C). The intensity profile of the two signals analyzed at the level of the white line (Figure 2C) highlights the complete overlapping of the two spectra, confirming the colocalization between α-synuclein and acetylated α-tubulin (Figure 2D). To validate our results, we repeated this staining using a panel of different antibodies, including three additional anti-α-synuclein antibodies combined with another anti-acetylated α-tubulin antibody (Appendix A), and we confirmed that α-synuclein and α-tubulin colocalize in post-mortem human brains.

These data indicate that the interplay between α-synuclein and acetylated α-tubulin occurs in post-mortem human brains.

### 2.3. α-Synuclein and Acetylated α-Tubulin Are in Close Proximity in Post-Mortem Human Brain

We then investigated the physical proximity of α-synuclein with acetylated α-tubulin taking advantage of the in situ Proximity Ligation Assay (PLA). This approach allows both the detection of protein–protein interaction directly on samples and the amplification of the signal [34]. With this strategy, we aimed to verify the α-synuclein/acetylated α-tubulin interplay directly on the post-mortem human brain and, at the same time, to amplify the faint signal of colocalizing α-synuclein previously observed (Figure 2B, arrows). As shown in Figure 3, the presence of green spots indicated that α-synuclein and acetylated α-tubulin were in close proximity along microtubules in the substantia nigra of the post-mortem human brain. The close proximity between α-synuclein and acetylated α-tubulin was confirmed by performing the PLA using a different couple of antibodies (Appendix A). Furthermore, we wondered whether this proximity was specific to dopaminergic neurons. Thus, we performed the PLA together with classical immunofluorescence for tyrosine hydroxylase (Appendix A), and we observed that the α-synuclein/acetylated α-tubulin PLA staining displayed a similar distribution in TH-positive and -negative neuropil (Appendix A). Interestingly, we quantified the PLA signal normalized on acetylated α-tubulin and observed a low but constant value (0.081 + 0.007; mean ± standard error of the mean) in the substantia nigra of the four post-mortem human brains.

These PLA results confirm the data emerging from the classical colocalization analysis and further support the occurrence of the physical interplay between α-synuclein and acetylated α-tubulin.

### 2.4. Increase of Acetylated α-Tubulin Affects α-Synuclein Oligomerization

Finally, we addressed whether the interplay between α-synuclein and acetylated α-tubulin affects α-synuclein aggregation. Recent data have indicated that alterations of the microtubule cytoskeleton could be involved in α-synuclein pathology [35]. α-synuclein aggregation is a multi-step process where, at the early stage, α-synuclein monomers join into small aggregates called oligomers [36]. To test the impact of α-tubulin acetylation on α-synuclein oligomerization, we used the selective HDAC6 inhibitor Tubacin to induce α-tubulin acetylation in a neuroblastoma SK-N-SH cell line overexpressing human wild-type α-synuclein (SK-N-SH Syn^+^, Appendix A, [37,38]). We evaluated α-synuclein aggregation with: (i) biochemical assays (western blotting analyses of differential soluble extracts); and (ii) the α-Synuclein PLA (AS-PLA) technique, which allows for oligomer detection and quantification in human tissues [39,40,41]. Once we verified that Tubacin treatment increased acetylated α-tubulin (Figure 4A and Appendix A), we analyzed Triton X-100 soluble and SDS-soluble fractions to evaluate the presence of different species of α-synuclein, i.e., monomeric, oligomeric, or aggregated α-synuclein (Appendix A). Monomeric α-synuclein was clearly detectable in the Triton X-100 soluble fraction and did not change following Tubacin treatment (Appendix A). On the contrary, the bands corresponding to oligomeric or aggregated species in the Triton X-100 soluble and SDS-soluble fractions were barely visible (Appendix A). Moving to the in situ detection of oligomeric α-synuclein using the AS-PLA (Figure 4A), we detected red puncta that were increased in the SK-N-SH Syn^+^ cells treated with Tubacin. The quantification of the area covered by AS-PLA puncta confirmed the increase in α-synuclein oligomer formation in the Tubacin-treated cells (Figure 4B), and this effect was dose-dependent (Appendix A). These observations support that an increase in α-tubulin acetylation can be involved in and promote α-synuclein aggregation.

In conclusion, with these experiments, we proved the existence of a link between the alteration of acetylated α-tubulin and the early step of α-synuclein aggregation.

## 3. Discussion

The acetylation of tubulin, along with the plethora of post-translational modifications that occur on α- and β-tubulin, strictly regulate microtubule behaviors, functions, and dysfunctions. Among the large amount of works investigating its role, recent evidence has pointed to its impact on the mechanical properties of microtubules [21,22]. However, the effect of α-tubulin acetylation on the interaction of the microtubule cytoskeleton with specific partners remains largely unexplored. To date, no studies have addressed the interplay between acetylated α-tubulin and α-synuclein, and this work is the first in this topic. We showed that α-synuclein and acetylated α-tubulin are strictly associated along microtubules in murine primary neurons and the human brain using advanced technology such as high-resolution microscopy and PLA. More intriguingly, we demonstrated that manipulating tubulin acetylation by the inhibition of HDAC6, the main α-tubulin deacetylase, promoted α-synuclein oligomer formation. These findings shed new light on the behaviors of α-synuclein in both the physiological and pathological context.

The interplay of α-synuclein with α-tubulin in the pre-synaptic compartment, where α-synuclein is more abundant, has been recently demonstrated [12]. In this study, we unraveled the ability of α-synuclein to interact with a specific subpopulation of microtubules, enriched by K40-acetylated α-tubulin, that is present in neurons mainly along the axon and dendrites [42]. This interplay could reveal a new convergence point between α-synuclein’s and microtubule cytoskeleton’s homeostasis. Indeed, both α-synuclein and acetylation of α-tubulin have been involved in common pathways. In neurons, acetylation of α-tubulin has been mainly linked to the regulation of morphogenesis, neuronal differentiation, and neuronal transport [23]. At the same time, α-synuclein has been demonstrated to regulate microtubule dynamics [6,7,8,27] at the basis of microtubule cytoskeleton formation and maintenance and, in turn, determine cellular morphogenesis [43] and intracellular transport [44,45]. Given that, the ability of α-synuclein to colocalize with acetylated microtubules and to be physically in close proximity with acetylated α-tubulin paves the way to further investigate how the microtubule cytoskeleton and α-synuclein work together to carry out their functions. Furthermore, the low but constant value of the α-synuclein/acetylated α-tubulin PLA signal on acetylated α-tubulin in the human brain could imply that its proper regulation is necessary in physiological conditions and, consequently, leads us to hypothesize that its alteration could be pathogenetic in accordance with emerging evidence [24].

The ability of α-synuclein to regulate microtubule dynamics has been investigated on total tubulin, independently from the presence of post-translational modifications. Indeed, microtubules are dynamic structures that continuously undergo cycles of growth and shortening in which soluble unpolymerized tubulin is attached to and detached from polymerized microtubules [46]. Based on the current knowledge, the acetylation of K40 on α-tubulin occurs mainly when it is incorporated into polymerized microtubules [47], while deacetylation affects soluble tubulin [48,49]. Previous data have shown that α-synuclein is able to interact with pre-formed and in vitro assembled microtubules [5], as well as with soluble tubulins [9,10]. Interestingly, its ability to interact and form a complex with free tubulin, or rather with a tetramer of α- and β-tubulins, can favor their binding to preformed microtubules, enhancing microtubules elongation [9]. Data emerging from this work show that α-synuclein is associated with a fraction of acetylated microtubules, thus suggesting that its impact on regulation of microtubule dynamics could be mediated by the presence of the acetyl group on the K40 residue of α-tubulin. Based on this, it could be important to verify the regulatory effects of α-synuclein on tubulin assembly and disassembly in vitro taking into account the potential contribution of tubulin acetylation.

Whether α-synuclein binds to the outer surface or the inner, the intralumenal surface of microtubules still needs to be determined. If the binding site is on the inner surface, the presence of a protein, although as small as α-synuclein is, could physically interfere with the action of the tubulin acetyl transferase αTAT [47]. Although several microtubule binding sites have been hypothesized on α-synuclein (see [4]), the findings remain unclear. As data from the literature indicate that α-synuclein interacts both with α- and β-tubulin [5,6,7,10,12,50,51], we hypothesize that this interaction could happen in the interface between α- and β-tubulins. This could fit with α-synuclein’s ability to form a complex with a tetramer of tubulins [9], which might involve two heterodimers that are assembled inside the single microtubule or belonging to parallel microtubules. The latter is in agreement with the presence of α-synuclein in between microtubules as we have shown in primary neurons.

Another hypothesis is that α-synuclein could bind acetylated α-tubulin inside the lumen of the microtubule. Indeed, αTAT is supposed to enter the microtubule through openings along the microtubule and to acetylate tubulin right there. By weakening inter-protofilament interactions, the acetylation of α-tubulin increases lattice plasticity, limits the spread of pre-existing lattice damage, and thus protects microtubules [21,22]. Stress-induced bending may cause transient openings that allow αTAT entry into the microtubule lumen in the portions subjected to stress [52]. The resulting increase in mechanical resilience could be a strategy for repairing the broken microtubules. We can speculate that α-synuclein could exert its function in this context, increasing and favoring microtubule suture, and thus justifying the interplay that we observed along the microtubule cytoskeleton.

Evidence supports the idea that an improper regulation of this post-translational modification affecting microtubules is involved in neurodegenerative processes [24]. The active form of HDAC6 is mainly sequestered in the Lewy bodies in dopaminergic neurons in post-mortem human brains [18], suggesting that the deacetylation of tubulin could be blocked or reduced in these cells and impact their fate. In this study, we found that HDAC6 deacetylase activity can influence the formation of α-synuclein oligomers, which are considered the earliest and most toxic species [53]. This is in accordance with the redistribution of acetylated α-tubulin during Lewy bodies’ morphogenesis observed in post-mortem human brains [54]. In line with a role of α-tubulin acetylation in α-synuclein aggregation, the inhibition of the tubulin deacetylase SIRT2 increases α-tubulin acetylation and reduces the formation of α-synuclein inclusion, finally alleviating PD-related neurotoxicity [55]. Here, we focused, for the first time, on the earliest step of α-synuclein aggregation that gives rise to the formation of α-synuclein oligomers. Furthermore, we inhibited the acetylase activity of HDAC6, and not SIRT2, known to deacetylate different subsets of acetylated microtubules [56]. The accumulation of acetylated α-tubulin, caused by the inhibition of HDAC6 deacetylase activity, induced an increase in α-synuclein oligomers, indicating that the alteration in this post-translational modification of the microtubule cytoskeleton could lead to the aggregation of α-synuclein and thus be implicated in PD pathology. Based on this, we believe that the two works are complementary and consolidate the role of acetylated α-tubulin regulation in α-synuclein pathology.

Whereas emerging evidence link HDAC6 to synucleinopathies [57,58], the impact of modulating its activity on neurodegeneration is still debated. To note, the inhibition of HDAC6 is considered a good strategy to contrast several neurodegenerative diseases [59], such as peripheral neuropathy [60,61,62], and, in particular, axon regeneration by increasing microtubule stability in the axonal shaft [63]. On the contrary, HDAC6 inhibition could be detrimental and contribute to the pathological deposition of protein aggregates that leads to neurodegeneration. Beyond regulating acetylation, HDAC6 plays a crucial role in protein homeostasis [64,65,66], as well as in aggresome formation and clearance of misfolded protein aggregates [17]. Thus, the overall inhibition of its activity can negatively impact the intracellular fate of the proteins, as shown in a previous study reporting that HDAC6 deficit led to the accumulation of α-synuclein into aggresome-like bodies [67]. In this context, our finding that HDAC6 inhibition triggers α-synuclein oligomerization is particularly relevant, as promoting HDAC6 activity during the initial steps of α-synuclein aggregation could counteract oligomer formation and exert a protective role. The identification of this time window can be crucial to use HDAC6 as future therapeutic target to prevent α-synuclein pathology.

In conclusion, this work highlights the physiological interplay between acetylation of α-tubulin and α-synuclein, suggesting that a proper regulation of acetylated tubulin is not only necessary to the proper function of microtubules but also for preventing the early step of α-synuclein aggregation. 

## 4. Materials and Methods

### 4.1. Animal

C57BL/6J wild-type (Charles River Laboratories Italia srl, Calco, Italy) mice were bred in the animal house facility at the Department of Bioscience of University of Milan, Milan, Italy. Animals were maintained in pathogen-free conditions, under a 12-h light-dark cycle at a room temperature (RT) of 22 °C, and had ad libitum food and water. All procedures were made in accordance to Italian law (D. Lgs n° 2014/26, implementation of the 2010/63/UE) and were approved by the University of Milan Animal Welfare Body and by the Italian Minister of Health (project code BDNS, n. 839C7.N.ARCH). 

### 4.2. Primary Cell Culture

Primary midbrain neurons were obtained from C57BL/6J wild-type mice at embryonic day 12.5 (E12.5) according to previously described protocols [68]. Briefly, midbrains dissected from E12.5 embryos were washed in HBSS (Euroclone, Pero, Italy) + 1% penicillin/streptomycin (Euroclone). After enzymatic digestion with Accumax (Sigma-Aldrich, St. Louis, MO, USA) and mechanical dissociation in the presence of 0.0625 mg/mL DNAse (Sigma-Aldrich), cells were resuspended in neurobasal medium (Thermo Fisher Scientific, Waltham, MA, USA) containing 1% penicillin/streptomycin (Euroclone), 1% L-Glutammine (Euroclone), and 2% B27 supplement (Thermo Fisher Scientific). Then, 150 cells/mm^2^ were seeded onto glass coverslips in 24 multi-well plates for immunofluorescence assays or directly on the well for western blot analyses. Both coverslips and wells were previously coated with 0.1 mg/mL poli-D-lysine (Sigma-Aldrich) and 0.01 mg/mL laminin (Sigma-Aldrich). Cells were maintained at 37 °C under a humidified atmosphere of 5% CO_2_ in the complete neurobasal medium for 7 days in vitro (DIV), changing half of the medium every two days. 

### 4.3. SK-N-SH Cell Culture

Human α-synuclein-stable transfected SK-N-SH (SK-N-SH Syn^+^) cells were routinely maintained in Dulbecco’s modified Eagle Medium Low Glucose (Sigma-Aldrich) in the presence of 1% L-Glutammine, 1% Non-Essential Amino Acids (Sigma-Aldrich), 1% penicillin/streptomycin, and 10% Fetal Bovine Serum (Euroclone). SK-N-SH Syn^+^ were plated in the presence of 50 µg/mL zeocin selection (Thermo Fisher Scientific). To induce differentiation, cells were seeded on 0.1 mg/mL Poli-D-Lysine-coated glass coverslips, treated with 3 µg/mL Retinoic Acid (Sigma-Aldrich) alone for 3 days and together with 80 nM TPA (Sigma-Aldrich) for an additional 3 days [69]. After 6 days of differentiation, cells were treated with DMSO (vehicle, Euroclone) and 6 µM or 12 µM Tubacin (Sigma-Aldrich) for 2 h at 37 °C, gently washed twice with Phosphate-Buffered Saline (PBS, Biosigma, Cona, Italy), and processed for analyses.

### 4.4. Human Brain Samples

Post-mortem human brains of healthy subjects (n = 4) were collected by the Nervous Tissues Bank (Milan, Italy). These subjects include three females (age at death: 82, 91, 93) and one male (age at death: 71). The absence of neurodegenerative disease was confirmed by the neuropathological analysis of autoptic brains, according to the BrainNet Europe Consortium guidelines [70]. In particular, the absence of incidental Lewy bodies was verified. The study procedures were in accordance with the principles outlined in the Declaration of Helsinki and approved by the Ethics Committee of University of Milan (protocol code 66/21), and a written informed consent was obtained from all the subjects before their enrollment.

Human samples from all subjects were obtained after autopsy and fixed in 10% buffered formalin for at least 21 days. After dehydration and clearing steps, the specimens containing substantia nigra were paraffin embedded and then cut in 5 μm-thick sections using a microtome (MR2258, Histoline, Pantigliate, Italy). Deparaffination and rehydration were performed before immunofluorescence and PLA experiments as previously described [18]. 

### 4.5. Immunofluorescence Assay

Primary cells and SK-N-SH Syn^+^ cells were analysed by immunofluorescence assays. Briefly, cells previously fixed with 4% paraformaldehyde + 10% glycerol, for 10 min at RT, were washed three times with PBS and then permeabilized with 1% Triton X-100 in PBS for 4 min at RT. After three washes with PBS, cells were incubated with saturation buffer (5% BSA in PBS) for 15 min at RT, and then with primary antibodies diluted in 1% BSA in PBS, for 1 h at 37 °C in a humified heater. The following antibodies were used for the immunofluorescence assay (Appendix A): anti-α-synuclein rabbit IgG (1:500; S3062, Sigma-Aldrich), anti-acetylated α-tubulin mouse IgG (1:300; clone 6-11 B-1, Sigma-Aldrich). After incubation, cells were washed three times and secondary Alexa-conjugated antibodies were added: Donkey anti-rabbit Alexa Fluor^TM^ 488 (1:1000; A21206, Invitrogen, Waltham, MA, USA); Donkey anti-mouse Alexa Fluor^TM^ 568 (1:1000; A10037, Invitrogen). Coverslips were incubated for 1 h at 37 °C in the dark and, after three washes with PBS, Hoechst 33342 (Invitrogen) was added (15 min at RT in the dark). Finally, coverslips were mounted with Mowiol-DABCO mounting medium. 

For human brains, substantia nigra sections were first incubated with 1% BSA and 0.1% Triton X-100 in PBS for 20 min at RT, and then with a mixture containing the following primary antibodies: anti-acetylated α-tubulin mouse IgG (1:300) or rabbit IgG (1:200, D20G3, Cell Signaling Technology, Danvers, MA, USA), anti-α-synuclein rabbit IgG (1:2000) or mouse IgG (1:500, Clone LB509, Abcam, Cambridge, UK; 1:1000, clone 4D6, Covance, Princeton, NJ, USA; 1:50, Syn-1, BD Biosciences, Franklin Lakes, NJ, USA), and anti-Tyrosine Hydroxylase goat IgG (1:200, PA5-18372, Invitrogen) overnight (ON) at RT. After three washes with PBS, a mixture of secondary antibodies was added for 2 h at RT in the dark. The secondary antibodies used were: Donkey anti-rabbit Alexa Fluor^TM^ 488, Donkey anti-mouse Alexa Fluor^TM^ 568, Donkey anti-mouse Alexa Fluor^TM^ plus 647, or Donkey anti-goat Brilliant Violet 421 (705-675-147, Jackson ImmunoResearch Europe LTD, Ely, UK). To visualize nuclei, Hoechst 33342 was added for 10 min at RT in the dark; finally, tissue sections were mounted with Mowiol-DABCO.

The specificity of the anti-α-synuclein antibody made in rabbit (S3062) used in this work was verified by pre-adsorption with the recombinant human α-synuclein protein. Detailed procedures are described in Appendix A, “Pre-adsorption of anti-α-synuclein S3062 antibody”, with relevant references [71,72].

### 4.6. Proximity Ligation Assay

Proximity Ligation Assay (PLA) was performed with the commercial Duolink^®^ kit (Sigma-Aldrich) according to the manufacturer’s instructions and carried out together with classical immunofluorescence. Briefly, to detect the close interaction between acetylated α-tubulin and α-synuclein, human brain sections were incubated with 1% BSA and 0.1% Triton X-100 in PBS for 1 h at RT and then with a mixture of primary antibodies containing anti-acetylated α-tubulin mouse antibody (1:150) and anti-α-synuclein rabbit antibody (1:1000), ON at RT. After three washes with PBS, Donkey anti-mouse MINUS probes and Donkey anti-rabbit PLUS probes (DUO92004; DUO92002, Sigma-Aldrich) in PLA diluent were added and incubated for 2 h at 37 °C, followed by three washes and incubation with ligase in Ligase solution for 1 h at 37 °C. Finally, Polimerase and Duolink reagent Green (Duolink^®^ In Situ Detection Reagents Green, DUO92014, Sigma-Aldrich) were added in a mix containing Donkey anti-mouse Alexa Fluor^TM^ 568 and Donkey anti-rabbit Alexa Fluor^TM^647 secondary antibodies and incubated for 2 h at 37 °C. Finally, Hoechst 33342 was added, for 10 min at RT in the dark, and sections were mounted with Mowiol-DABCO. To evaluate the close proximity between α-tubulin and α-synuclein in dopaminergic neurons, the PLA experiment was repeated, adding the anti-tyrosine hydroxylase antibody (1:200) in the primary antibody mixture ON at RT. In parallel, to validate the obtained results, the experiment was performed using anti-acetylated α-tubulin rabbit antibody (1:100) and anti-α-synuclein mouse antibody (1:250, clone LB509). After incubation with the MINUS and PLUS probes, samples were incubated with a mixture containing Polimerase and Duolink reagent Red (Duolink^®^ In situ Detection Reagents Red, DUO92008 Sigma-Aldrich) together with Donkey anti-goat Brilliant Violet 425, Donkey anti-rabbit Alexa Fluor^TM^ 488 and Donkey anti-mouse Alexa Fluor^TM^ 647 secondary antibodies. Finally, sections were mounted with Mowiol-DABCO.

To detect the presence of α-synuclein oligomers in SK-N-SK Syn^+^ cells, PLUS and MINUS probes (DUO92009 and DUO92010, Sigma-Aldrich) were directly conjugated to α-synuclein primary antibodies according to the protocols previously described [39,41]. In details, cells were incubated with a mixture containing α-synuclein S3062-MINUS and α-synuclein S3062-PLUS probes, as well as anti-acetylated α-tubulin antibody, in PLA Diluent for 90 min at 37 °C. After three washes with PBS, Duolink ligation mix containing Ligase was added, and cells were incubated for 1 h at 37 °C. After three washes, a mixture containing Duolink amplification reagent Red (Duolink^®^ In Situ Detection Reagents Red, DUO92008, Sigma-Aldrich), Polymerase, and anti-mouse Alexa Fluor^®^ 488 (715-545-151; Jackson Immuno Research Europe LTD) secondary antibody was added and incubated for 100 min at 37 °C in the dark. Finally, after incubation with Hoechst 33342, coverslips were mounted with Mowiol-DABCO mounting medium. Finally, the presence of α-synuclein oligomers was also investigated by Western blotting according to the procedures described in Appendix A, “Biochemical fractionation”, with relevant references [73,74].

### 4.7. Confocal Microscopy

For cell acquisition, images were collected at 120× magnification (1024 × 1024) with an oil-immersion 60× objective or at 100× magnification (1024 × 1024) with an oil-immersion 100× objective using a Nikon (Tokyo, Japan) Ai-SIM laser scanning confocal microscope. For each primary midbrain culture preparation, images containing neurons were acquired and analyzed. For the Manders coefficient analyses, each single z-stack was analyzed. For PLA analyses, the maximum projection of each image was analyzed to quantify AS-PLA puncta. Analyses of confocal images were performed with FiJi software (NIH, Bethesda, MD, USA). In detail, the JaCoP plugin was used to measure the colocalization and Manders coefficients, while “analyze particles” was used for the AS-PLA puncta analyses. For super-resolution images, the N-SIM E structured light super-resolution module was used coupled with a 100× oil-immersion objectives (512 × 512). Human brain sections were analyzed with a Nikon spinning disk confocal microscope, and the images were collected at 60× magnification (2048 × 2048) with an oil-immersion 60× objective or at 100× magnification (2048 × 2048) using a 100× silicone-immersion objectives. 

### 4.8. Statistical Analyses

The statistical analyses were conducted using GraphPad Prism 8 software (San Diego, CA, USA) and a paired *t*-test. A *p* value < 0.05 was considered statistically significant. 

## Figures and Tables

**Figure 1 ijms-24-12287-f001:**
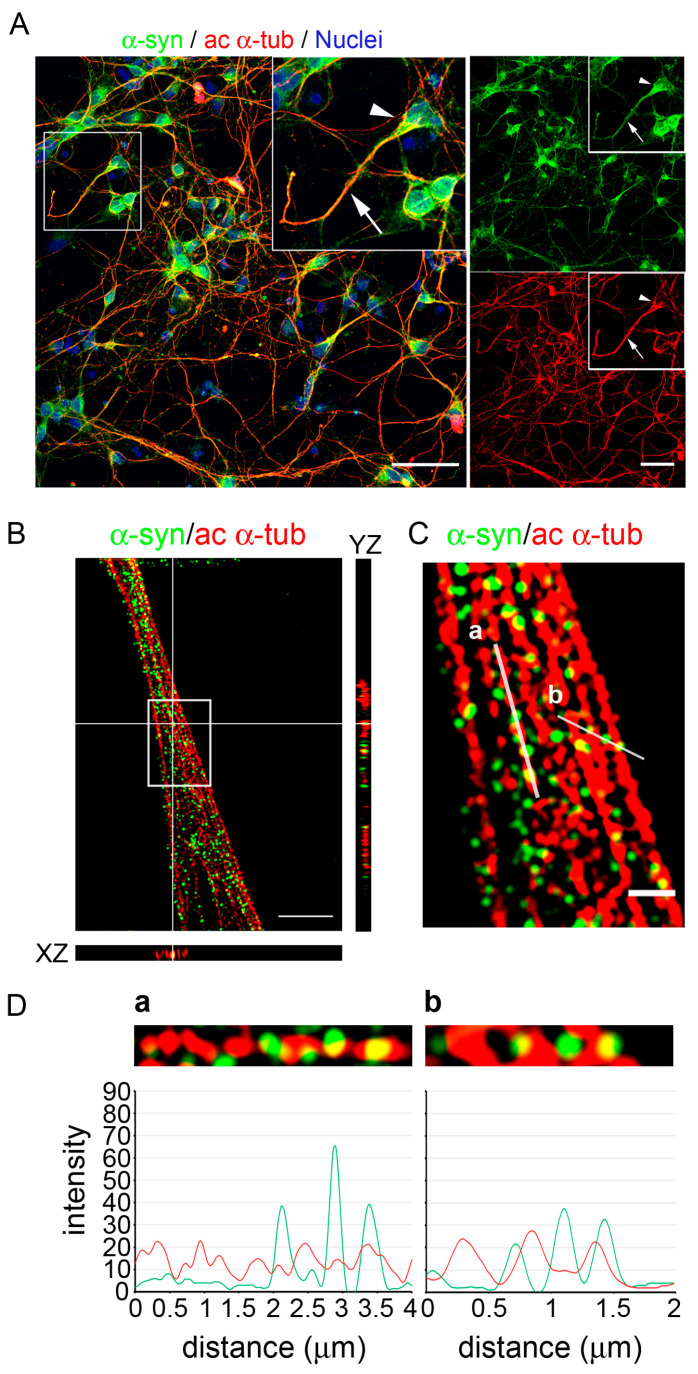
α-synuclein colocalized with acetylated α-tubulin along the microtubules in the primary midbrain neurons. (**A**) Confocal microscopy image shows the distribution of α-synuclein (α-syn, in green) and acetylated α-tubulin (ac α-tub, in red). The white square indicates the position of the 2× zoomed inset, in which acetylated α-tubulin staining is clearly detectable in a single neuron along a neurite (arrow) but less evident in cellular body (arrowhead). α-synuclein was visible both in the cell bodies and along neurites. Nuclei were counterstained with Hoechst. Scale bar: 50 μm. (**B**) Super-resolution confocal image (Structured Illumination Microscopy, SIM) and orthogonal view showing the distribution of α-synuclein and acetylated α-tubulin along a bundle of microtubules. The orthogonal projections of the merged channels represent the XZ (bottom) and YZ (right) planes and highlight the partial colocalization of α-synuclein with acetylated α-tubulin. White square: inset shown in panel (**C**). Scale bar: 5 µm. (**C**) Merge image representing a single reconstructed image plane of α-synuclein and acetylated α-tubulin signals. Scale bar: 1 µm. (**D**) Intensity profile for each channel using the same color code of SIM image, representing (**a**) the portion of a single microtubule and (**b**) a portion of three parallel microtubules, as indicated by the white lines in (**C**).

**Figure 2 ijms-24-12287-f002:**
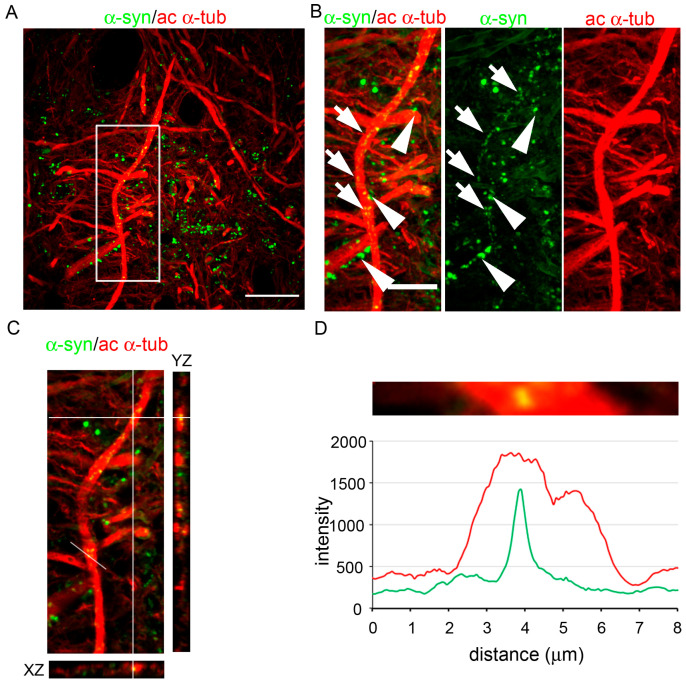
α-synuclein colocalized with acetylated α-tubulin along microtubules in the post-mortem human brain. (**A**) Confocal microscopy analysis showing the distribution of α-synuclein (α-syn, in green) and acetylated α-tubulin (ac α-tub, in red) in the substantia nigra. Scale bar: 20 μm. (**B**) Zoomed detail of the merged and the single channels of the inset in (**A**). The big and intense α-synuclein staining (arrowheads) and the small and faint one (arrows) are visible in the images. Scale bar: 10 μm. (**C**) The orthogonal projections of the merged channels represent XZ (bottom) and YZ (right) planes and highlight the partial colocalization of α-synuclein with acetylated α-tubulin. (**D**) Detail of a single optical section showing a portion of acetylated microtubules in which α-synuclein is present (white line in (**C**)) and the relative intensity profile of the two signals.

**Figure 3 ijms-24-12287-f003:**
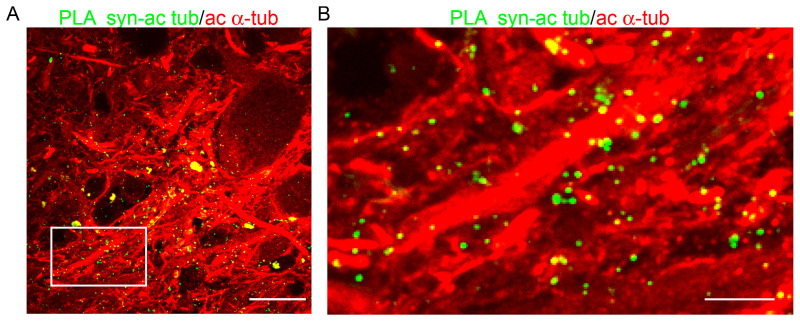
α-synuclein and acetylated α-tubulin are in close proximity in the post-mortem human brain. (**A**) Representative confocal image of the substantia nigra of the post-mortem human brain showing the presence of the in situ Proximity Ligation Assay (PLA) signal indicating the close interplay between α-synuclein and acetylated α-tubulin (PLA syn-ac tub, in green). Scale bar: 20 µm. (**B**) Zoomed detail of the inset in A. PLA syn-ac tub staining is clearly detectable along acetylated microtubules. Scale bar: 5 µm.

**Figure 4 ijms-24-12287-f004:**
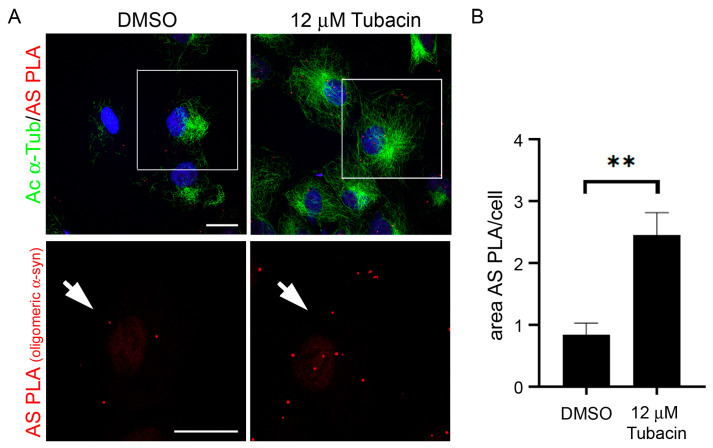
The increase of tubulin acetylation induced α-synuclein oligomers’ formation. (**A**) Acetylated α-tubulin (in green) and oligomeric α-synuclein (AS-PLA, in red) in SK-N-SH Syn^+^ treated for 2 h with 12 µM Tubacin compared to DMSO control. The lower panels show the 2× magnified view of selected squared areas. Arrow: AS-PLA staining indicating oligomeric α-synuclein. Scale bar: 20 µm. (**B**) Graph represents the mean area of the oligomeric α-synuclein in the analyzed cells (expressed as area AS PLA/cell). N = 4 biological replicates, ** *p* = 0.0080, according to the paired *t*-test.

## Data Availability

The datasets used and analyzed during the current study are available from the corresponding authors upon reasonable request.

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
