# Peer review of "Acetylated α-Tubulin and α-Synuclein: Physiological Interplay and Contribution to α-Synuclein Oligomerization"

_ijms, 2023, doi:10.3390/ijms241512287_

Round 1

Reviewer 1 Report

In this article, Calogero and colleagues studied the interplay between α-synuclein and acetylated α-tubulin, and the putative effects of the modulation of tubulin acetylation on its colocalization with α-syn, as well as the oligomerization of the latter. Authors found a colocalization between α-syn and ac-α-tubulin in mouse primary neurons and human Parkinson’s Disease patient brain sections using immunofluorescence microscopy. They also observed the proximity of α-syn and ac-α-tubulin using Proximity Ligation Assay, and showed that the increase of acetylation of the latter induced an increase of proximity (termed oligomers) of α-syn in cultured SK-N-SH cells overexpressing human α-syn.

Overall, the manuscript is very well written, clear, and easy to understand. Microscopic images are of a good quality and authors do not overstate their findings. Also, the Supplementary figures, including uncropped images and sup data is appreciable. 

However, I raised several technical and scientific concerns that substantially limit the relevance of the study in the field.

1- The choice of (and limitation to one single) α-syn antibody.

Calogero and colleagues used in all IF, PLA, and blots a single α-syn antibody, that is not frequently used in the field. I would highly recommend to validate the findings with other antibodies such as Syn-1 (BD-bioscience), D37A6 (Cell Signalling) or MJFR-1(Abcam) for the detection of total, murine, and human α-syn respectively. As a matter of fact, the Western blots of SK-N-SH with the S3062 Sigma antibody shows a double band in the WT cells, as well as unspecific bands (on Sigma datasheets). As authors base all findings of IF, PLA etc on the use of this antibody, they have to make sure the protein stained is really α-syn.

2- The relevance of colocalization of these two proteins

Authors showed colocalization measurements of α-syn and ac-α-tubulin on IF microscopic images. As these two proteins are highly abundant, I am skeptical on the relevance of a localized partial colocalization of these two proteins. Also, authors show this colocalization on a PD patient brain section, but they must show a side-by-side control healthy subject section to see if the disease condition is increasing or decreasing the proximity of these two proteins.

Even in this case, the study is limited to a measurement of the physical proximity of two protein components.

3- Alpha-synuclein oligomerization

Even in the case where their antibody is efficiently (and uniquely) binding to α-syn in their oligomerization assay (I recommend here the use of human-specific MJFR-1 antibody), there is still a possibility that the increased proximity of α-syn proteins (I do not mention oligomerization on purpose) could be due to an effect of Tubacin itself. Also, the link between α-syn oligomers formation, and its pathological amyloid aggregation is worlds apart. 

Authors would have to perform biochemistry experiments, seeing oligomers in western blots, or with crosslinking. They should even do differential centrifugations on cell lysates, or insolubility assays in order to identify the formation of characterized oligomeric assemblies, rather than a “simple” proximity observation.

Reviewer 2 Report

In this manuscript, the authors investigated the association between α-synuclein and microtubules in mouse brain cells and post-mortem human brain samples. They successfully demonstrated the close proximity of α-synuclein and acetylated α-tubulin. Additionally, the authors explored the impact of α-tubulin acetylation on α-synuclein oligomerization and observed an increase in oligomer formation with modification of α-tubulin acetylation levels using Tubacin. These findings suggest a potential role of α-tubulin acetylation in the early stages of α-synuclein aggregation and propose it as a therapeutic target for Parkinson's disease. The comprehensive analysis from the cellular level to the clinical level greatly contributes to the physiological significance of the study, making it suitable for publication in IJMS.

Minor points:

(1) In Figures 1, 2, and 3 (or if the autopsy brain is too valuable to use, only Figure 1 is available), please include the number of cells analyzed, denoted as n = x, to indicate that the observed phenomenon is not limited to the representative figures.

(2) In the immunohistochemical analysis of the autopsied brain shown in Figure 3, it is apparent that the stained cells are tubulin+ neurons. However, it is unclear whether they are dopaminergic neurons or if Lewy bodies are present. Since there are other types of neurons in the midbrain, it would be valuable to discuss the selectivity of dopaminergic neurons. Including an image stained with TH (tyrosine hydroxylase) would provide a clearer understanding.
